# COMPASSDOCK 🧭: COMPREHENSIVE ACCURATE ASSESSMENT APPROACH FOR DEEP LEARNING-BASED MOLECULAR DOCKING IN INFERENCE AND FINE-TUNING

## ABSTRACT

Datasets used for molecular docking, such as PDBBind, contain technical variability - they are noisy. Although the origins of the noise have been discussed, a comprehensive analysis of the physical, chemical, and bioactivity characteristics of the datasets is still lacking. To address this gap, we introduce the Comprehensive Accurate Assessment (Compass). Compass integrates two key components: PoseCheck, which examines ligand strain energy, protein-ligand steric clashes, and interactions, and AA-Score, a new empirical scoring function for calculating binding affinity energy. Together, these form a unified workflow that assesses both the physical/chemical properties and bioactivity favorability of ligands and protein-ligand interactions. Our analysis of the PDBBind dataset using Compass reveals substantial noise in the ground truth data. Additionally, we propose CompassDock, which incorporates the Compass module with DiffDock, the state-of-the-art deep learning-based molecular docking method, to enable accurate assessment of docked ligands during inference. Finally, we present a new paradigm for enhancing molecular docking model performance by fine-tuning with Compass Scores, which encompass binding affinity energy, strain energy, and the number of steric clashes identified by Compass. Our results show that, while fine-tuning without Compass improves the percentage of docked poses with RMSD < 2Å, it leads to a decrease in physical/chemical and bioactivity favorability. In contrast, fine-tuning with Compass shows a limited improvement in RMSD < 2Å but enhances the physical/chemical and bioactivity favorability of the ligand conformation. The source code is available at `https://github.com/anonym8171iclr2025/iclr_2025_paperid_8171`.

## 1 INTRODUCTION

Molecular docking is crucial in drug discovery as it helps determine the feasibility of potential drug candidates. Traditional docking methods are slow (Trott & Olson, 2010; Halgren et al., 2004), leading to the development of deep learning (DL)-based methods (Stärk et al., 2022; Lu et al., 2022). These newer approaches have shown impressive accuracy and significantly faster performance, decreasing the run time from hours/days to seconds/minutes per protein-ligand pair (Corso et al., 2022; 2024; Lu et al., 2024).

Deep learning-based methods typically assess their accuracy using numerical metrics such as Root Mean Square Distance (RMSD) (Meli & Biggin, 2020). Although RMSD measures the overall deviation of individual molecules from the ground truth, it is a purely distance-based metric and does not consider the physico-chemical interactions or bioactivity properties between the ligand and the protein at the binding site. As a result, metrics like RMSD fail to capture critical factors that determine strong molecular binding, such as the favorability of these interactions. PoseBuster (Buttenschoen et al., 2024) demonstrated that even when structures have RMSD < 2Å, it is common to find ligands with unfavorable physico-chemical properties.

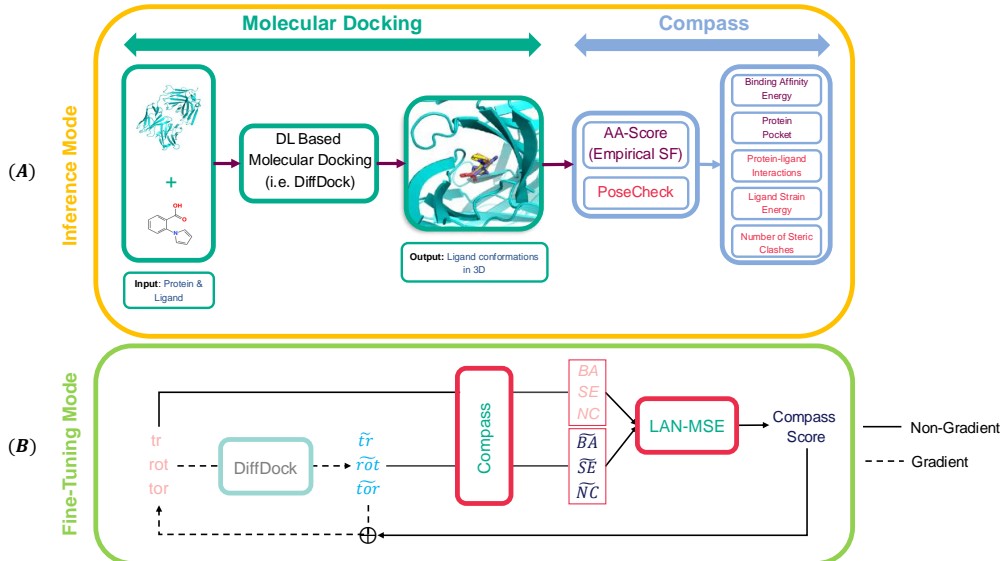

Figure 1: **(A) Inference Mode:** Provides comprehensive PCB feature quality assessment during DL-Based Molecular Docking Inference runtime using integrated Compass workflow. **(B) Fine-Tuning Mode:** Introduces a new fine-tuning method for DL-Based Molecular Docking with a PCB-based approach, utilizing the Integrated Compass Score.

PoseCheck (Harris et al., 2023) was developed for Structure-Based Drug Design (SBDD) (Blundell, 1996; Ferreira et al., 2015; Anderson, 2003) to analyze molecular stability by calculating strain energy, identifying interactions within the complex, and counting steric clashes between the protein and ligand. Another critical metric for assessing protein-ligand interactions is binding affinity energy, which reflects bioactivity. Autodock-Vina (Trott & Olson, 2010) is a widely used method for calculating binding affinity energy. However, the recently introduced AA-Score (Pan et al., 2022), an empirical scoring function, has been shown to be more accurate than Autodock-Vina. Additionally, AA-Score provides the 3D structure of the docked ligand within the protein pocket (pdb file), allowing for easier visual inspection of docking results.

Additionally, (Corso et al., 2024) demonstrated that model fine-tuning boosts performance, yielding an increased percentage of structures' with an RMSD score of less than 2Å. While fine-tuning without Physico-Chemical and Bioactivity (PCB) features increases the percentage of docked poses with RMSD < 2Å, we demonstrate that this approach results in degraded physico-chemical interactions and bioactivity. On the other hand, fine-tuning with PCB results in only a slight improvement in RMSD < 2Å but enhances the ligands' physico-chemical and bioactivity favorability.

To tackle the challenges in DL-based molecular docking, such as identifying ligand strain energy, counting protein-ligand steric clashes, finding interaction types, finding the best score function for binding affinity energy, and enhancing performance with a PCB-based approach, we propose CompassDock with two modes (Figure 1):

- **Inference Mode:** Provides comprehensive PCB quality analysis during DL-based molecular docking with integrated Comprehensive Accurate Assessment (Compass) module.

- **Fine-Tuning Mode:** Introduces a new paradigm for fine-tuning DL-based molecular docking using a PCB-based approach with the Integrated Compass Score.

Our analysis of the PDBBind (Liu et al., 2017) dataset, commonly used in molecular docking, revealed significant PCB noise/unfavorability. Secondly, we give users the opportunity to analyze their docking results with real assessments in an inference run. Finally, with these PCB-based assessments, we can improve the model by fine-tuning it with a new score function called Compass Score.

In summary, our CompassDock offers the following key contributions:

- A unified module, named Compass, was developed to determine PCB features of docked molecular conformations sampled by DL-based molecular docking during inference mode.

- A PCB quality/favorability analysis of why DL methods do not achieve the desired performance, highlighting the noisy nature of the training dataset (i.e. PDBBind).

- Proposing a novel loss function called Log Absolute Normalized - Mean Square Error (LAN-MSE) which prevents exploding loss values and effectively reduces the impact of outliers thereby enhancing model robustness.

- A new objective function, called Compass Score, which incorporates binding affinity energy, strain energy, and steric clashes into LAN-MSE.

- A new paradigm introduced to enhance molecular docking model performance in PCB quality by fine-tuning DL-based docking models using Compass Score.

## 2 METHODS

The proposed CompassDock method not only enables accurate and realistic PCB quality assessments in DL-based molecular docking during inference, such as with DiffDock but also enhances the model's ability to capture favorable PCB features by incorporating these properties into the fine-tuning process using a novel score (Compass Score) and loss function (LAN-MSE). The overall structure of CompassDock is illustrated in Figure 1.

### 2.1 MOTIVATION

Generally, the performance of the docking methods is evaluated using the RMSD metric, where RMSD values below 2 Ångstroms(Å) are considered nearly accurate (Alhossary et al., 2015; Hassan et al., 2017; McNutt et al., 2021). However, PoseBuster (Buttenschoen et al., 2024) demonstrated that even in structures with RMSD < 2Å, ligands can still display unfavorable physicochemical properties. We further illustrated this by incorporating a bioactivity score during CompassDock inference, as shown in Figure 2, in order to highlight that the RMSD metric does not necessarily lead to physicochemically and bioactively favorable conformation structures. Despite an RMSD of 1.23 within the same protein pocket region, the sampled molecular conformation exhibits less favorable binding affinity energy, along with higher strain energy and steric clashes compared to the ground truth of PDB ID 1a46.

Also, when we have new protein-ligand pairs, we don't always have Ground Truth Data. DiffDock Confidence Score (Corso et al., 2022) is used to rank the various conformations, but what we discussed in RMSD also applies to the Confidence Score.

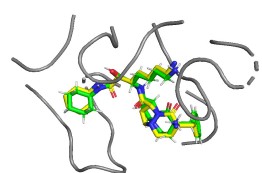

**PDB ID: 1a46, RMSD: 1.23**

| **PDB ID**: 1a46 | Gr. Tr. | Inf. Pred. |
|---|---|---|
| Bind. Aff. | -6.46 | -3.13 |
| Str. En. | 0.16 | 11.9 |
| # of Clash. | 3 | 19 |

Figure 2: Example of RMSD Fails. **Green**: Ground Truth 1a46 Ligand's Conformation; **Yellow**: Inference Prediction of 1a46 Ligand's Conformation; **Gray**: Ground Truth & Inference Prediction of 1a46 Protein Pocket .

To accurately evaluate protein-ligand molecular docking, a unified workflow integrated with DL-based docking methods, such as DiffDock, is needed to provide information on binding affinity, strain energy, steric clashes, and interaction types. To achieve this, we propose CompassDock, which utilizes the Compass module, combining PoseCheck and AA-Score, without the need for complex package management.

Despite advancements in deep learning (DL)-based molecular docking methods, their performance improvements remain limited, partly due to the noise present in the PDBBind dataset used for training, which has not been thoroughly discussed or analyzed. An examination of approximately 14,000 protein-ligand pairs in the PDBBind dataset using Compass reveals the presence of unfavorable PCB values, as shown in Figure 3.

Finally, (Corso et al., 2024) showed that fine-tuning the model can improve the percentage of structures with RMSD $< 2$Å. However, while fine-tuning without incorporating PCB features increases the proportion of docked poses achieving RMSD $< 2$Å, our results demonstrated a decrease in the favorability of PCB features. To improve favorable PCB features, we introduce a new regularizer called **Compass Score**, which penalizes the loss function during fine-tuning based on PCB features.

## 2.2 COMPASSDOCK

As shown in Figure 1, CompassDock operates in two modes: Inference Mode (Section 2.3) and Fine-Tuning Mode(Section 2.4). In both modes, the core component is a unified module called Compass, which integrates with DL-based molecular docking methods, such as DiffDock, to assess how well molecular conformations exhibit favorable PCB features, including ligand strain energy, binding affinity energy, and the number of steric clashes between protein-ligand pairs. The Compass workflow/module consists of two key elements: PoseCheck and AA-Score. All source codes are available [1].

### 2.2.1 DIFFDOCK

DiffDock (Corso et al., 2022; 2024) approaches molecular docking as a problem of learning a distribution of ligand poses based on the protein structure, utilizing a diffusion generative model (DGM) developed over this defined space. While (De Bortoli et al., 2022) initially described DGMs on submanifolds by projecting a diffusion from ambient space onto the submanifold, they refined this by mapping to a more manageable manifold where the diffusion kernel is directly sampled, thereby enhancing the efficiency of training the DGM on this new manifold.

Ligand movements are mapped in molecular docking to mathematical groups, associating translations with the 3D translation group $T(3)$, rigid rotations with the 3D rotation group $SO(3)$, and changes in torsion angles with multiple copies of the 2D rotation group $SO(2)$. These movements are formalized by defining how each group acts on a ligand pose $c \in \mathbb{R}^{3n}$, where translations are straightforward additions to atom positions, and rotations are conducted around the ligand's center of mass. Furthermore, modifications to torsion angles are defined to minimally perturb the structure in an RMSD sense and can be specified separately from translations and rotations to ensure changes are disentangled.

Overall, DiffDock consists of two primary models: the Score Model and the Confidence Model.

**Score Model** $s(x, y, t)$ processes the current ligand pose $x$ and protein structure $y$ in 3D space, outputting in the tangent space $T_r T^3 \oplus T_R \mathrm{SO}(3) \oplus T_\theta \mathrm{SO}(2)^m$. It generates two SE(3)-equivariant vectors for translations and rotations, along with an SE(3)-invariant scalar for each rotatable bond, utilizing architectures based on SE(3)-equivariant convolutional networks over point clouds (Thomas et al., 2018; Geiger et al.). Additionally, the Score Model represents structures as heterogeneous geometric graphs, incorporating features derived from protein sequences and convolutions specific to translation, rotation, and torsion to optimize for multiscale integration and computational efficiency (Lin et al., 2022; Jing et al., 2022).

**Confidence Model** $d(x, y)$, on the other hand, takes the same inputs of ligand pose and protein structure and outputs a single scalar that is SE(3)-invariant, reflecting the joint rototranslations of $x$ and $y$. This model focuses on assessing the stability and plausibility of the ligand pose relative to the protein structure, providing a scalar output that can guide the refinement of docking predictions.

### 2.2.2 POSECHECK

PoseCheck (Harris et al., 2023) is developed to assess molecular stability by measuring strain energy, identifying interactions within the complex, and counting steric clashes between the protein and ligand.

**Strain Energy**: Strain energy is the internal energy accumulated in a ligand due to conformational changes during binding, affecting bond lengths, angles, and torsional stability, which in turn influences the binding affinity and stability of the protein-ligand complex (Perola & Charifson, 2004).

---

[1]https://github.com/anonym8171iclr2025/iclr_2025_paperid_8171

Lower strain energy typically leads to more favorable binding interactions and could enhance therapeutic effectiveness due to the balance between enthalpy and entropy. The calculation of strain energy involves determining the difference in internal energy between a relaxed pose and the generated pose using the Universal Force Field (UFF) in RDKit (Rappé et al., 1992).

**Steric Clashes**: Steric clashes occur when two neutral atoms are closer than the sum of their van der Waals radii, indicating an energetically unfavorable and physically implausible interaction (Ramachandran et al., 2011; Buonfiglio et al., 2015). Such clashes suggest that the current conformation of the ligand within the protein is suboptimal, highlighting potential inadequacies in pose design or a fundamental mismatch in molecular topology. In SBDD, the total number of these clashes is considered a crucial performance metric, with a clash defined as the pairwise distance between protein and ligand atoms falling below their combined van der Waals radii by a tolerance of 0.5 Å (Harris et al., 2023).

**Interaction Fingerprint**: Interaction fingerprinting is a computational technique in SBDD that captures and analyzes interactions between a ligand and its target protein, encoding these interactions as a bit vector known as an interaction fingerprint (Bouysset & Fiorucci, 2021; Marcou & Rognan, 2007). This method allows for the rapid comparison of different ligands or binding poses by calculating the Tanimoto similarity between fingerprints, providing a quantitative measure of interaction similarity. The approach uses specific libraries like ProLIF to encode molecular interactions like hydrogen bonds and hydrophobic contacts into a compact, easily comparable format (Bouysset & Fiorucci, 2021).

### 2.2.3 AA-Score

A new empirical scoring function by (Pan et al., 2022), named AA-Score, which includes several amino acid-specific interaction components such as hydrogen bonds, electrostatic, and van der Waals interactions (Appendix B.1). This scoring function differentiates interactions with main-chain and side-chain atoms across these types and incorporates additional energy components that are not amino acid-specific, including hydrophobic contacts, $\pi$-$\pi$ stacking, $\pi$-cation interactions, metal-ligand interactions, and a conformational entropy penalty. Each of these energy components is thoroughly described to provide a comprehensive understanding of their contributions to the scoring function.

### 2.3 INFERENCE MODE

In inference mode, the protein (provided as either a sequence or 3D structure) and the ligand (given as a SMILES string or 3D structure) are input into DiffDock to generate 3D ligand conformations. Compass then analyzes the protein and ligand 3D structures to evaluate binding affinity, strain energy, the number of steric clashes, and protein interaction types. Additionally, we developed a recursive redocking method to further optimize the molecular conformation during inference, making the PCB features' quality more favorable. This process is described by the following Equation 1:

$$f(L_i, P, n) = \begin{cases} L_i & \text{if the conformation is favorable} \\ \text{stop} & \text{if the conformation is unfavorable} \\ \text{stop} & \text{if } n \geq N_{\max} \\ f(L_{i+1}, P, n+1) & \text{otherwise (further refinement is needed)} \end{cases} \tag{1}$$

**where:**

- $L_i$: The current ligand conformation at step $i$,
- $P$: The protein structure,
- $f$: The recursive function that refines ligand conformations based on physico-chemical and bioactivity (PCB) features,
- $n$: The current iteration step,
- $N_{\max}$: The maximum number of iterations allowed.

This function iterates through docking steps, checking each conformation against the thresholds. It halts the recursion when the optimal conditions are met, when the scores are too low, or when the maximum iteration limit $N_{\max}$ is reached. If none of these conditions are satisfied, it continues to the next conformation $L_{i+1}$, refining the results until a favorable state is achieved or the iteration limit is reached.

## 2.4 FINE-TUNING MODE

### 2.4.1 LOSS REGULARIZER WITH COMPASS SCORE FOR FINE-TUNING

Fine-tuning of the DL-based docking method focuses on reducing the overall structural deviation of individual molecules from the ground truth but tends to overlook important PCB features' quality of ligands and protein-ligand interactions. To address this, we used Compass to fine-tune the model with the goal of improving the quality of these PCB features. To more effectively compare these results with ground-truth data, we propose replacing traditional regression methods with a new loss function, Log Absolute Normalized - Mean Square Error (LAN-MSE), which better handles the normalization of outliers (Section 2.4.2).

### 2.4.2 LOG ABSOLUTE NORMALIZED - MEAN SQUARE ERROR (LAN-MSE)

The LAN-MSE loss function applies a logarithmic transformation to both predicted and true values before computing the mean squared error (MSE). This logarithmic approach effectively reduces the impact of outliers, enhancing model robustness by prioritizing relative rather than absolute differences. This characteristic is particularly advantageous in scenarios where output values span large scales.

**Theorem: Stability of Logarithmic Transformation.** *Let $x$ be a real number. The logarithmic function $\log(|x| + buffer)$ is stabilized by a buffer $buffer = 1.1$ for $|x| < 1$ to prevent the function from diverging or becoming overly sensitive near zero.*

**Proof:** For $x$ approaching zero, $\log(|x|)$ approaches $-\infty$. Introducing a buffer shifts the domain away from zero, ensuring that $\log(|x| + buffer)$ remains finite and non-sensitive as $x \to 0$.

Error normalization in LAN-MSE utilizes the term $2|\log(|y_i| + \text{buffer}_{\text{true},i})|$, scaling each error according to the magnitude of the true value. This scaling mechanism proportionately decreases the significance of errors for larger true values, aligning the error metric with scenarios where percentage differences outweigh absolute differences.

LAN-MSE is ideally suited for applications involving data with exponential growth or decay, as it adjusts for the logarithmic nature of such data. The following function provides a mathematical representation of LAN-MSE, which includes mechanisms for handling the peculiarities of the data:

$$\text{LAN-MSE}(\hat{y}, y) = \frac{1}{N} \sum_{i=1}^{N} \left( \frac{\log(|y_i| + \text{buffer}_{\text{true},i}) - \log(|\hat{y}_i| + \text{buffer}_{\text{pred},i})}{2|\log(|y_i| + \text{buffer}_{\text{true},i})| + \epsilon} \right)^2 \qquad (2)$$

**where:**

- $\hat{y}_i$ represents the predicted value for the $i$-th data point.

- $y_i$ is the true value for the $i$-th data point.

- $\text{buffer}_{\text{true},i}$ is set to 1.1 if $|y_i| < 1$, otherwise, it is 1.0.

- $\text{buffer}_{\text{pred},i}$ is similarly set based on the magnitude of $\hat{y}_i$.

- $\epsilon$ is a small constant (e.g., $10^{-5}$) to prevent division by zero in the denominator.

This detailed definition underscores how the application of buffers and normalization constants in the loss function ensures a nuanced evaluation of prediction accuracy across various data magnitudes. (Details for further Analysis in Appendix C)

### 2.4.3 COMPASS SCORE

The Compass Score is a key element of the Fine-Tuning mode in CompassDock. It compares the PCB features (binding affinity, strain energy, and number of steric clashes) of the sampled molecular conformation with those of the ground truth conformation using the LAN-MSE loss function.

**Regularizers for Compass Score:**

- **Binding Affinity Regularizer:** Binding affinity is calculated based on the translational ($tr$), rotational ($rot$), and torsional ($tor$) movements and is compared to the ground truth using the LAN-MSE loss function:

$$\text{Compass Score}_{\text{Binding Affinity}} = \text{LAN-MSE}(\hat{y}_{\text{Binding Affinity}}, y_{\text{Binding Affinity}}) \quad (3)$$

where $\hat{y}_{\text{Binding Affinity}}$ represents the predicted binding affinity energy, and $y_{\text{Binding Affinity}}$ denotes the true binding affinity value.

- **Strain Energy Regularizer:** Strain energy is calculated based on the same movements and compared to the ground truth strain energy using the LAN-MSE loss function:

$$\text{Compass Score}_{\text{Strain Energy}} = \text{LAN-MSE}(\hat{y}_{\text{Strain Energy}}, y_{\text{Strain Energy}}) \quad (4)$$

- **Steric Clash Regularizer:** The number of steric clashes is calculated based on the same movements and compared to the ground truth using LAN-MSE:

$$\text{Compass Score}_{\text{Steric Clash}} = \text{LAN-MSE}(\hat{y}_{\text{Steric Clash}}, y_{\text{Steric Clash}}) \quad (5)$$

Given the uncertain relative importance of these three physical properties, they are assumed to contribute equally to the overall Compass Score:

$$\text{Compass Score}_{\text{Total}} = \frac{\mathcal{CS}_{\text{Binding Affinity}} + \mathcal{CS}_{\text{Strain Energy}} + \mathcal{CS}_{\text{Steric Clashes}}}{3} \quad (6)$$

where $\mathcal{CS}_{\text{Binding Affinity}}$ denotes Compass Score$_{\text{Binding Affinity}}$ illustrated in Equation 3; $\mathcal{CS}_{\text{Strain Energy}}$ denotes Compass Score$_{\text{Strain Energy}}$ illustrated in Equation 4; $\mathcal{CS}_{\text{Steric Clashes}}$ denotes Compass Score$_{\text{Steric Clashes}}$ illustrated in Equation 5. This score acts as a Non-Gradient-Tracked Loss (or loss function penalizer) and does indirectly influence parameter updates through gradients during the fine-tuning process.

### 2.4.4 LOSS FUNCTION

To optimize CompassDock in Fine-Tuning mode, the overall loss function incorporates the total Compass Score, shown in Equation 6, which acts as a non-gradient-tracked penalizer alongside the primary DiffDock loss, with both components weighted accordingly:

$$\mathcal{L}_{\text{Total}} = \mathcal{L}_{\text{DiffDock}} \cdot w_{\text{DiffDock}} + \mathcal{CS}_{\text{Total}} \cdot w_{\text{Compass Score}} \quad (7)$$

where $\mathcal{L}_{\text{DiffDock}}$ represents the DiffDock loss function, which updates translation, rotation, and torsional angles; $w_{\text{DiffDock}}$ is the weight assigned to the DiffDock loss; $\mathcal{CS}_{\text{Total}}$ represents the total Compass Score (as defined in Equation 6); $w_{\text{Compass Score}}$ is the weight assigned to the total Compass Score, calculated as $1 - w_{\text{DiffDock}}$.

## 3 EXPERIMENTS

### 3.1 PCB QUALITY ANALYSIS

The experiments used the PDBBind dataset, specifically the preprocessed version applied by Diff-Dock. In experiments, the unified Compass workflow was employed to evaluate binding affinity, strain energy, and the number of clashes. A notable portion of the preprocessed ligands were found to have unfavorable PCB features.

The dataset comprised approximately 14,000 protein-ligand pairs. The remaining 5,000 pairs, preprocessed by EquiBind (Stärk et al., 2022), were excluded from the analysis due to errors in bond structures detected (Details in Appendix A.1) by PoseCheck (Harris et al., 2023) and AA-Score (Pan et al., 2022).

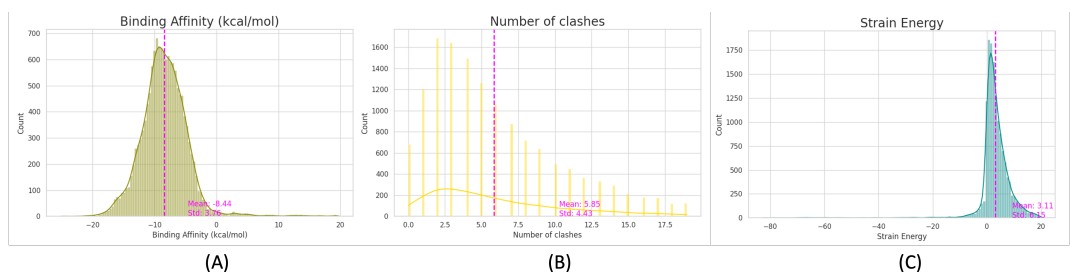

Figure 3: Distribution of PCB properties and quality within the PDBBind ground truth dataset. **(A):** Distribution of binding affinity across the PDBBind dataset. **(B):** Distribution of the number of steric clashes between protein-ligand pairs in PDBBind. **(C):** Distribution of strain energy in ligands from the PDBBind dataset.

## 3.2 FINE-TUNING WITH COMPASS SCORE PENALIZER

Instead of fine-tuning the entire dataset, an initial subset of 900 protein-ligand pairs from the PDB-Bind validation set was selected. Due to structural issues in some of the preprocessed data, as discussed in Section 3.1 (Appendix A.1), this subset was further reduced to approximately 700 samples. Given the relatively long computation time for the Compass Score, 80 protein-ligand pairs were randomly selected for training and 20 for validation.

For the test set, similar structural issues resulted in the selection of about 261 samples from the preprocessed data.

### 3.2.1 EXPERIMENT SETTINGS

During the fine-tuning, the weights for the DiffDock loss function ($w_{\text{DiffDock}}$) were set at 0.99, 0.9, and 0.75 (Section 2.4.4). The corresponding weights for the Compass Score ($w_{\text{Compass Score}}$) were determined as $1 - w_{\text{DiffDock}}$.

Additionally, the hyperparameters from the DiffDock-L model were applied, and various learning rates—0.1, 0.01, 0.001, and 0.0001—were used to optimize performance.

## 4 RESULTS

### 4.1 PCB QUALITY ANALYSIS

We analyzed the PDBBind ground truth data using the unified Compass workflow (Section 3.1), focusing on binding affinity, steric clashes between protein-ligand pairs, and ligand strain energy, as illustrated in Figure 3.

Figure 3A shows the distribution of binding affinity in the PDBBind dataset, with a mean of -8.44 kcal/mol and a standard deviation of 3.76 kcal/mol. In Figure 3B, the average number of steric clashes is 5.85, with a standard deviation of 4.43; notably, only about 5% of the dataset has no steric clashes, indicating minimal physical violations. Finally, Figure 3C illustrates the distribution of ligand strain energy, with a mean of 3.11 and a standard deviation of 6.15.

### 4.2 FINE-TUNING WITH COMPASS SCORE PENALIZER

The performance of CompassDock in Fine-Tuning mode was benchmarked (Section 3.2) against DiffDock-L in inference and DiffDock with fine-tuning based on the percentage of ligands with RMSD < 2Å and RMSD < 5Å, as well as the percentage of ligands with favorable PCB features (binding affinity < 0, strain energy < 5, and number of steric clashes < 5), as shown in Figure 4.

The baseline DiffDock-L model, consisting of 30 million parameters, achieved an accuracy of 11.38% for RMSD below 2 Å and 24.67% for RMSD below 5 Å on the selected 261-sample test set (Appendix A.1). After fine-tuning, the DiffDock-L model improved slightly, reaching 11.57%

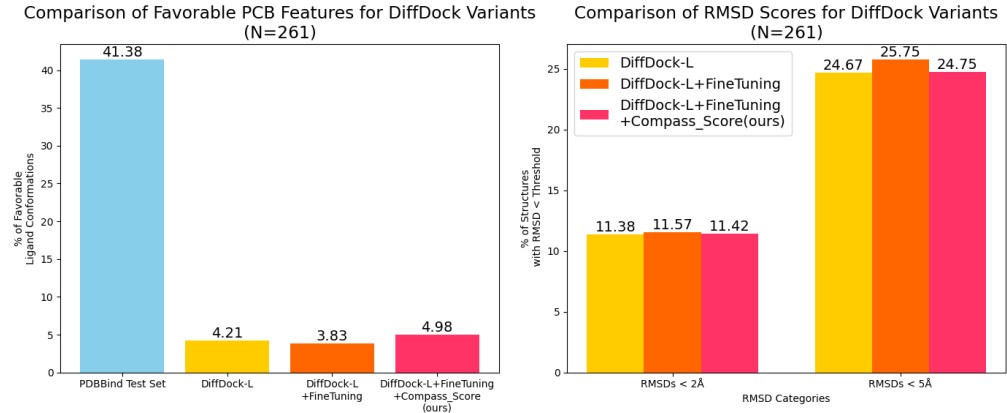

Figure 4: Benchmarking DiffDock models: pre-trained model (DiffDock-L), fine-tuned model (DiffDock-L+FineTuning), and fine-tuning with Compass Score (CompassDock (ours)), based on RMSD and favorable PCB properties.

| | 1st epoch | 33rd epoch | 53rd epoch | 55th epoch |
|---|---|---|---|---|
| Binding Affinity (-9.59) | 2139.99 | 625.09 | 14.68 | -5.08 |
| Strain Energy (3.92) | 1.93 | 0.12 | 1.91 | 1.94 |
| # of Clash (14) | 145 | 150 | 22 | 5 |
| RMSD | 7.77 | 8.25 | 6.17 | 5.92 |

Figure 5: Comparison of PDB ID: 5c28 Ligand Conformation predictions during fine-tuning. **Green Structures**: **Ground Truth Conformation of Ligand**; **Cyan Structures**: **Ground Truth Structures of Binding Pocket**. Ground Truth Binding Affinity, Strain Energy, and Number of Clashes indicated as green in the first column

accuracy for RMSD below 2 Å and 25.75% for RMSD below 5 Å. Applying the Compass Score method yielded a modest enhancement, with 11.42% accuracy for RMSD below 2 Å and 24.75% for RMSD below 5 Å.

In terms of favorable PCB ligands (defined as binding affinity < 0, strain energy < 5, and number of steric clashes < 5), the original PDBBind test set achieved 41.38%. The DiffDock-L model performed at 4.21%, and fine-tuning further reduced this to 3.83%. However, applying the Compass Score in CompassDock improved the percentage of favorable PCB ligand conformations to 4.98%.

## 5 DISCUSSION

This study conducted two key experiments to assess the effectiveness of the Compass Score within the molecular docking framework using the PDBBind dataset. The first experiment (Section 4.1) evaluated PCB feature quality, revealing notable variations in binding affinity, steric clashes, and strain energy, which highlight the complexities and inconsistencies of protein-ligand interactions. The findings showed that most of the dataset exhibited steric clashes, with only a small portion free from such violations. This suggests that even widely-used datasets like PDBBind contain inherent inaccuracies that could influence docking prediction outcomes. These results emphasize the need for robust modeling techniques that can handle the diverse physicochemical and biochemical properties

found in various ligand-protein pairs, underscoring the challenges in predicting accurate docking conformations. This aligns with the approach taken by the PLINDER dataset (Durairaj et al., 2024).

Although the use of the Compass Score as a loss penalizer (Section 4.2) resulted in only a slight improvement in RMSD accuracy compared to standard fine-tuning, our results showed that standard fine-tuning tends to reduce the capture of favorable PCB properties. In contrast, incorporating the Compass Score improved the favorable PCB characteristics. This highlights the effectiveness of integrating PCB properties into the fine-tuning process of molecular docking algorithms.

Furthermore, as shown in Figure 5, RMSDs greater than 5 Å can still result in viable molecular conformations. While lower RMSDs correlate (Alhossary et al., 2015; Hassan et al., 2017; McNutt et al., 2021) with improvements in binding affinity, strain energy, or steric clashes, it is not accurate to assert that a specific RMSD threshold consistently leads to better molecular conformations during docking processes. This point is further exemplified by the RMSD discrepancies shown in Figure 6, where despite an RMSD value of 4.75 for the 2si0 PDB ID ligand in the initial fine-tuning epoch, significant violations in PCB are evident, highlighting the RMSD's limitations.

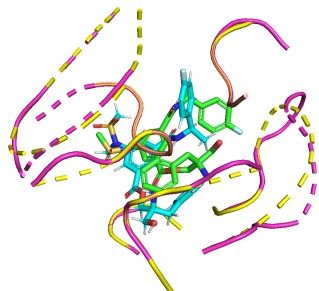

| **PDB ID**: 2is0 | Ground Truth | Predicted [2] |
|---|---|---|
| Binding Affinity | -11.33 | 3505.32 |
| Strain Energy | 7.31 | 20.65 |
| Number of Clashes | 6 | 205 |

Figure 6: Example of Difference of PCB Properties of Ground Truth and Predicted Sample during the early stages of Fine-Tuning and its visualization. RMSD value is 4.75. **Yellow**: **Ground Truth** Protein Pocket[3]; **Purple**: **Predicted** Protein Pocked; **Green**: **Predicted** Conformation of Ligand; **Cyan**: **Ground Truth** Conformation of Ligand

## 6 CONCLUSION

These findings support the integration of detailed PCB features into docking algorithms to improve predictive accuracy and assessment. Future research should investigate the scalability of refined scoring mechanisms across larger datasets and explore the incorporation of more complex interaction dynamics to further advance DL-based molecular docking. This approach holds the potential to significantly enhance the predictive capabilities of computational models, leading to more accurate outcomes in drug discovery and biomolecular research.

---

[2]This is not final result of the ligand. Selected first epoch of fine-tuning.

[3]Ground Truth Pocket determined by Compass' AA-Score Module. Same for Predicted samples. See the details in Figure 1

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

## A EXPREIMENT DETAILS

### A.1 DATASET

In the PDBBind dataset, a subset of the ligands' bonds was preprocessed or encoded in a manner that rendered them incompatible with the AA-Score and PoseCheck modules. Specifically, AA-Score was unable to process 125 protein-ligand pairs, while PoseCheck failed to process 3,040 pairs. Despite these challenges, Compass successfully ran analyses on 16,070 protein-ligand pairs.

Given the presence of significant outliers within the dataset, which could skew the distribution analysis, we applied a filtering criterion. We selected only those protein-ligand pairs exhibiting values less than 20 in Binding Affinity, Strain Energy, and Clashes to ensure more meaningful distribution analysis. This filtering process resulted in a refined dataset comprising approximately 13,977 protein-ligand pairs for subsequent analysis.

## B METHOD DETAILS

### B.1 AA-SCORE

AA-score (Pan et al., 2022) calculates the binding affinity energy as illustrated in Equation 8:

$$\Delta G_{\text{binding}} = \sum_i \Delta G_{\text{hb-main}}^{AA} + \sum_i \Delta G_{\text{hb-side}}^{AA} + \sum_i \Delta G_{\text{ele-main}}^{AA} + \sum_i \Delta G_{\text{ele-side}}^{AA} +$$
$$\sum_i \Delta G_{\text{vdW-side}}^{AA} + \Delta G_{\text{hc}} + \Delta G_{\text{stacking}} + \Delta G_{\text{cation}} + \Delta G_{\text{metal}} + \Delta G_{\text{rot}} + \tag{8}$$
$$\sum_i \Delta G_{\text{vdW-main}}^{AA}$$

Table 1 summarizes energy components like Hydrogen Bond, van der Waals interaction, Electrostatic interaction, Hydrophobic contacts, $\pi - \pi$ interaction, $\pi$-cation interaction and Metal-Ligand Interactions.

| Category | Number of Components | Description |
|---|---|---|
| Hydrogen bond | 32 | $\Delta G_{\text{hb-main}}^{AA^i}, \Delta G_{\text{hb-side}}^{AA^i}$ |
| vdW interaction | 40 | $\Delta G_{\text{vdW-main}}^{AA^i}, \Delta G_{\text{vdW-side}}^{AA^i}$ |
| Electrostatic interaction | 80 | $\Delta G_{\text{ele-main}}^{AA^i}, \Delta G_{\text{ele-side}}^{AA^i}$ |
| Hydrophobic contacts | 1 | $\Delta G_{\text{hc}}$ |
| $\pi-\pi$ | 1 | $\Delta G_{\pi-\pi}$ |
| $\pi$-cation | 1 | $\Delta G_{\pi-\text{cation}}$ |
| Metal–ligand interaction | 1 | $\Delta G_{\text{metal}}$ |
| Number of rotatable bonds | 1 | $\Delta G_{\text{rot}}$ |

Table 1: Description of energy components in the AA-Score

#### B.1.1 HYDROGEN BOND INTERACTION

The contribution of hydrogen bonds to overall energy depends on the bond's length between the donor and acceptor atoms, and the angle formed with the hydrogen atom. (Pan et al., 2022) defined a hydrogen bond based on two criteria: the bond length must not exceed 3.5 Å, and the angle must be at least 120°. The strength of each hydrogen bond is calculated as following Equation 9 by (Yan et al., 2017):

$$\Delta G_{\text{hbond}}^{AA^i} = W_i^{hb} \sum_m^{AA} \sum_n^{lig} \left( \frac{1}{1 + \left(\frac{d_{mn}}{2.6}\right)^6} / 0.58 \right) \tag{9}$$

where $\Delta G_{\text{hbond}}^{AA^i}$ represents the energy contribution from the hydrogen bond interaction between atom $m$ in the ligand and atom $n$ of amino acid type $i$ within the binding pocket. The term $d_{mn}$ denotes the distance between the donor and acceptor atoms involved in the hydrogen bonding interaction.

### B.1.2 HYDROPHOBIC CONTACTS

The hydrophobic interactions are quantified by summing the contributions of all hydrophobic atom pairs between the protein and the ligand. This calculation follows the methodology established by ChemScore (Eldridge et al., 1997; Murray et al., 1998), whereby the hydrophobic interaction energy is computed based on the specific pairing criteria as shown in Equation 10:

$$\Delta G_{\text{hc}}^{AA^i} = W_{hc}^i \sum_m^{AA} \sum_n^{lig} f(d_{mn}) \tag{10}$$

where

$$f(d) = \begin{cases} 1.0 & \text{if } d \le d_0 + 0.5 \\ \frac{1}{1.5} \times (d_0 + 2.0 - d) \times d_0 + 0.5 & \text{if } d_0 + 0.5 < d \le d_0 + 2.0 \\ 0 & \text{if } d > d_0 + 2.0 \end{cases} \tag{11}$$

where $\Delta G_{\text{hc}}^{AA^i}$ represents the hydrophobic interaction energy between amino acid type $i$ and the ligand, calculated by summing the contributions of all hydrophobic matches between the ligand and protein amino acid type $i$. The term $d_0$ is the sum of the atomic radii of atom $m$ and atom $n$, and $d_{mn}$ denotes the distance between ligand atom $m$ and protein atom $n$ of amino acid type $i$.

### B.1.3 ELECTROSTATIC INTERACTION

The electrostatic interaction is quantified using the Coulomb's law as follows:

$$\Delta G_{\text{ele}}^{AA^i} = W_{\text{ele}}^i \sum_m^{AA} \sum_n^{lig} \frac{q_m \times q_n}{d_{mn}} \tag{12}$$

where $\Delta G_{\text{ele}}^{AA^i}$ signifies the electrostatic interaction energy between atom $n$ in the ligand and atom $m$ from amino acid type $i$ in the binding site. Here, $q_m$ and $q_n$ are the partial charges on atoms $m$ and $n$, respectively, and $d_{mn}$ denotes the distance between these atoms.

### B.1.4 VAN DER WAALS INTERACTIONS

AA-Score employed a modified Lennard-Jones potential for computing the van der Waals (vdW) interactions as following:

$$\Delta G_{\text{vdW}}^{AA^i} = W_{\text{vdW}}^i \sum_m^{AA} \sum_n^{lig} \left[ \left( \frac{d_0}{d_{mn}} \right)^8 - 2 \left( \frac{d_0}{d_{mn}} \right)^4 \right] \tag{13}$$

where $\Delta G_{\text{vdW}}^{AA^i}$ quantifies the energy contribution from the van der Waals interaction between amino acid type $i$ and the ligand. This interaction is derived by aggregating the effects from all atom pairs between the ligand atoms and the protein atoms of amino acid type $i$. The term $d_{mn}$ denotes the atomic distance between the ligand atom $m$ and protein atom $n$ of amino acid type $i$, while $d_0$ is the sum of the atomic radii $r_m$ and $r_n$, used in AA-Score calculations (Wang et al., 2002).

### B.1.5 METAL-LIGAND INTERACTION

Protein binding sites often include metal ions such as $Zn^{2+}$, $Cu^{2+}$, $Fe^{3+}$, among others, which are crucial for the stability of protein-ligand complexes. It is utilized from (Wang et al., 1998) where they calculate the metal-ligand binding as the following Equation 14

$$\Delta G_{\text{metal}} = W_{\text{metal}} \sum_{m}^{lig} \sum_{n}^{metal} f(d_{mn}) \tag{14}$$

where

$$f(d) = \begin{cases} 1.0 & \text{if } d < 2.0 \\ 3.0 - d & \text{if } 2.0 \leq d \leq 3.0 \\ 0.0 & \text{if } d > 3.0 \end{cases} \tag{15}$$

where, $\Delta G_{\text{metal}}$ quantifies the energy from metal-ligand interactions, where $m$ is the coordinating atom in the ligand, and $d_{mn}$ is the distance between ligand atom $m$ and the metal atom $n$.

### B.1.6 $\pi$-$\pi$ INTERACTION.

$\pi$-stacking interactions occur when the distance between the centers of two aromatic rings is less than 5.5 Å. The angle between the normals of the rings should deviate no more than 30° from the optimal orientation (0° for face-to-face and 90° for edge-to-face configurations) (Salentin et al., 2015; de Freitas & Schapira, 2017). The interaction strength is computed in Equation 16:

$$\Delta G_{\pi - \pi} = W_{\pi - \pi} \sum_{m}^{prot} \sum_{n}^{lig} f(d_{mn}, \theta, \sigma) \tag{16}$$

where

$$f(d, \theta, \sigma) = \begin{cases} 1 & \text{if } 60° \leq \theta < 90° \text{ and } 0.5 \leq d \leq 5.5 \text{ and } \sigma \leq 2 \\ 1 & \text{if } \theta \leq 30° \text{ or } \theta \geq 150° \text{ and } 0.5 \leq d \leq 5.5 \text{ and } \sigma \leq 2 \\ 0 & \text{otherwise} \end{cases} \tag{17}$$

Here, $d$ represents the distance between the centers of the aromatic rings, $\theta$ is the angle between the normals to these rings, and $\sigma$ is the perpendicular distance between one ring center and the plane of the other ring.

### B.1.7 $\pi$-CATION INTERACTION.

The $\pi$-cation interaction is considered significant when the distance between the center of an aromatic ring and a charged group is less than 5.5 Å. It is illustrated in Equation 18:

$$\Delta G_{\pi - \text{cation}} = W_{\pi - \text{cation}} \sum_{m}^{prot} \sum_{n}^{lig} f(d_{mn}) \tag{18}$$

where

$$f(d) = \begin{cases} 1 & \text{if } 0.5 \leq d \leq 5.5 \\ 0 & \text{otherwise} \end{cases} \tag{19}$$

here $d$ is the distance between the center of the aromatic ring and the center of the cation.

## C THEORETICAL ANALYSIS OF LAN-MSE

### C.1 MATHEMATICAL JUSTIFICATION FOR LOGARITHMIC TRANSFORMATION

The utilization of logarithmic transformations in the LAN-MSE is substantiated through several mathematical properties:

- **Dimensionless Measurement:** Logarithmic transformations convert scale-dependent quantities into dimensionless measures, facilitating comparisons across data of different scales.

- **Relative Error Sensitivity:** By employing logarithms, the LAN-MSE accentuates sensitivity to relative errors over absolute differences, aligning it with scenarios where proportional discrepancies are more significant than absolute magnitudes.

Consider two sets of values $(y_i, \hat{y}_i)$ and their scaled versions $(ky_i, k\hat{y}_i)$, where $k$ is a positive scalar. We aim to demonstrate that the LAN-MSE remains invariant under scaling of the input data.

The logarithmic transformation used in the LAN-MSE scales as follows when the data are multiplied by a scalar $k$:

$$\log(|ky_i| + \text{buffer}) = \log(k(|y_i| + \frac{\text{buffer}}{k})) \tag{20}$$

Using the property that $\log(ab) = \log(a) + \log(b)$, we can express this as:

$$\log(k) + \log(|y_i| + \frac{\text{buffer}}{k}) \tag{21}$$

As $k$ is constant across all terms in the LAN-MSE, it will factor out in the subtraction within the MSE computation:

$$\log(|y_i| + \text{buffer}_{\text{true},i}) - \log(|\hat{y}_i| + \text{buffer}_{\text{pred},i}) = \log(k) + \log(|y_i| + \frac{\text{buffer}_{\text{true},i}}{k}) - \\ \left(\log(k) + \log(|\hat{y}_i| + \frac{\text{buffer}_{\text{pred},i}}{k})\right) \tag{22}$$

Simplifying, the $\log(k)$ terms cancel each other out:

$$\log(|y_i| + \frac{\text{buffer}_{\text{true},i}}{k}) - \log(|\hat{y}_i| + \frac{\text{buffer}_{\text{pred},i}}{k})$$

Hence, the LAN-MSE computed for the original and scaled values yields the same result:

$$\text{LAN-MSE}(\hat{y}, y) = \text{LAN-MSE}(k\hat{y}, ky)$$

This demonstrates the scale invariance of the LAN-MSE, which is essential in fields where data magnitudes can span multiple scales, ensuring that the loss function treats all data uniformly regardless of their absolute magnitude.

## C.2   STABILITY ANALYSIS

The stabilization of the logarithmic function via a buffer is examined by considering the limit behavior as $x$ approaches zero:

$$\lim_{x \to 0^+} \log(|x| + \text{buffer}) = \log(\text{buffer}) \tag{23}$$

This limit is finite and well-defined, ensuring that the logarithmic function remains stable and less sensitive to small perturbations in $x$.

## C.3    IMPACT OF THE BUFFER ON ERROR DYNAMICS

The buffer's influence on the error dynamics is reflected in the derivative of the logarithmic function:

$$\frac{d}{dx}\log(|x| + \text{buffer}) = \frac{1}{|x| + \text{buffer}} \tag{24}$$

This derivative indicates reduced sensitivity as $|x|$ becomes very small, resulting in less aggressive penalization for small deviations in $x$, which can be beneficial when dealing with noisy data or outliers.

## C.4    NORMALIZATION AND SCALING

The normalization using $2|\log(|y_i| + \text{buffer}_{\text{true},i})|$ not only scales the errors but adapts the error metric to the magnitude of the true values:

$$\text{Normalized Error} = \frac{\log(|y_i| + \text{buffer}_{\text{true},i}) - \log(|\hat{y}_i| + \text{buffer}_{\text{pred},i})}{2|\log(|y_i| + \text{buffer}_{\text{true},i})| + \epsilon} \tag{25}$$

This adaptive scaling ensures that each data point's error is balanced proportionally, allowing for a more equitable influence across the dataset.

## C.5    PRACTICAL IMPLICATIONS AND LIMITATIONS

While the LAN-MSE provides numerous advantages, it may introduce biases in datasets where zero or negative values are significant, as the logarithmic function requires positive arguments. Moreover, the sensitivity of model performance to the parameters $\epsilon$ and buffer values necessitates careful tuning, possibly supported by empirical testing or sensitivity analysis.

