# OpenReview forum: "CompassDock: Comprehensive Accurate Assessment Approach for Deep Learning-Based Molecular Docking in Inference and Fine-Tuning"
_ICLR.cc/2025/Conference — ICLR 2025 Conference Withdrawn Submission_

### Official Review · Reviewer_jJXN · 2024-11-02

**Soundness:** 3
**Presentation:** 1
**Contribution:** 2
**Rating:** 3
**Confidence:** 4

**Summary:**

The paper proposes a new framework to select poses from diffusion-based molecular docking models or finetune them in such a way that they optimize the energetic characteristics of the resulting poses.

**Strengths:**

Although the paper inherits ideas from previous works proposing finetuning methods or pose quality evaluation techniques for docking models, the idea of combining these into one is novel to my knowledge and has the potential to be an important component to improve these models.

**Weaknesses:**

Unfortunately, I believe that the paper requires significant rewriting to make the methodology clear and the state of the experimental validation seems too preliminary for such a venue.

In terms of the writing the whole method section is largely unclear but below I will highlight some of the critical components:

1. Equation 1: the first two options (favorable and unfavorable) seem to cover all possibilities so I don’t see the use for the others, I suspect the second option is wrong. Moreover, what does “stop” mean in this case? Which pose will be returned?
2. The LAN-MSE loss is very confusing. Firstly is y always positive or negative? Otherwise, I don’t understand how taking the absolute value should be a good idea given that this potentially incentivizes the prediction and true value to diverge to the positive and negative counterparts, which for example with the binding affinity my understanding is that this would be very bad.
3. Still on the LAN-MSE, the definition of buffer, is confusing, from the “where” in Eq 2 it seems to suggest that buffer, should simply be written as a function of |y| instead of as a constant. Moreover, the resulting definition of f(|y|) = |y|+bugger(|y|) that is used after the log for the regression is a discontinuous function with f(0.99) > f(1), a property that seems clearly detrimental.
4. I would not call the statement on the stability of log a theorem.
5. What does it mean for the binding affinity, strain energy, steric clash to be “calculated based on the translational, rotational and torsional movements”? (near Eq 3)
6. What is a “Non-Gradient-Tracked-Loss”? (line 351)
7. I don’t understand how Eq 7 is optimized. How is the CS_total score used as a differentiable loss during finetuning of a diffusion model? Two issues that the authors do not explain how they go around (1) how they train the per step diffusion process on a loss only computable on endstate (2) how to train a model like DiffDock operating on submanifold with a loss differentiable, I assume, only in Euclidean space.

Results:

8. Results on the inference usage of CompassDock do not seem to be provided.
9. Why is the DiffDock performance so much lower on this version of the PDBBind test set compared to the traditionally used test set?
10. There does not seem to be any significant improvement in performance from the finetuning (the reported numbers are 4.21% to 4.98% and 11.38% to 11.42%). Can the authors point to stronger evidence for the benefit of their framework? Does further scaling the finetuning to more structure start providing significant improvement?

**Questions:**

See above

---

### Official Review · Reviewer_qTG1 · 2024-11-02

**Soundness:** 1
**Presentation:** 2
**Contribution:** 2
**Rating:** 3
**Confidence:** 4

**Summary:**

This paper describes CompassDock, which combines the previously developed PoseCheck and AA-score evaluation strategies, and applies this to evaluate the results of docking with DiffDock and fine tune DiffDock.  Scoring components from PoseCheck/AA-score are combined using a log-based transform, LAN-MSE.

**Strengths:**

Incorporating stronger physical priors about ligand quality is a reasonable approach to improving docking quality.

**Weaknesses:**

CompassDock inference mode is described, but not evaluated.  I was expecting an evaluation of DiffDock where DiffDock's confidence model is replaced by CompassScore.

Fine-tuning is done by adding the CompassScore as a "non-gradient-tracked penalizer" to the loss.  This raises the question of how the CompassScore could possibly influence the model weights given it does not contribute to the gradients in any way (reviewing the provided source code confirms this is what is happening).

The most likely explanation for the observed improvements in fine-tuning, especially given the small training/test set size, is that they are statistical artifacts of the stochastic nature of the training.  Not assessment of statistical significance is provided.

If the score is affecting fine-tuning in someway I am missing, I would expect a more comprehensive evaluation on the improvement in the components of the CompassScore (e.g. predicting binding affinity should improve significantly). However, only anecdotal evidence of this is provided.

**Questions:**

What is the point of inference mode? What evaluations can you provide that indicate its value?

Can you explain how the CompassScore is affecting fine-tuning?

Are the observed results statistically significant across multiple training runs?

---

### Official Review · Reviewer_vEQR · 2024-11-03

**Soundness:** 1
**Presentation:** 2
**Contribution:** 1
**Rating:** 1
**Confidence:** 5

**Summary:**

This paper presents an approach for fine-tuning pre-trained deep learning models used in molecular docking. The proposed method introduces a unified module called Compass, which integrates PoseCheck, a tool for examining the physicochemical and biological properties of docking poses, and AA-score, an empirical scoring function for protein-ligand interactions. Additionally, the authors propose a loss function named Log Absolute Normalized Mean Square Error (LAN-MSE) to enhance the robustness of the fine-tuning process.

**Strengths:**

The study presents a new paradigm for enhancing molecular docking model performance by fine-tuning with Compass Scores, which encompass binding affinity energy, strain energy, and the number of steric clashes identified by Compass.

**Weaknesses:**

This paper does not adequately support its key claim mentioned in the abstract: improving the physicochemical and biological favorability of conformations is beneficial. Additionally, the tool used to evaluate the quality of structures, PoseCheck, is problematic. As a reviewer, I examined its code for detecting steric clashes and found that it uses element-based van der Waals radii, which is overly simplistic for crystal structures. This approach is not precise enough to reliably determine whether a crystal structure has favorable physicochemical and biological properties. Moreover, the use of empirical scoring functions is also insufficiently considered, as it does not account for the discrepancy between the predicted binding scores and the actual experimental values.

**Questions:**

1.	The fine-tuned model proposed by the authors does not demonstrate any meaningful improvement in the docking task; it could be even problematic.
2.	The authors repeatedly claim that the conformations generated by the fine-tuned model exhibit better physicochemical and biological properties, but they only provide a few examples without conducting a large-scale evaluation of its performance.
3.	The authors argue that many of the crystal structures in existing datasets have unreasonable conformations, but this conclusion is based solely on PoseCheck, which is an insufficient argument. Furthermore, PoseCheck itself has not been proven as a reliable tool for assessing the quality of crystal structures.
4.	When the authors introduced the empirical AA-score as a loss function for fine-tuning, they did not address how the relationships between the predicted conformation’s AA-score/the crystal structure’s AA-score and the actual binding affinity were handled. According to the original AA-score paper, its correlation between the predicted binding affinity and the experimental values for crystal structures is relatively low.
5.	The fine-tuning implemented in the authors' code is actually ineffective because the Compass loss, as written, behaves as a constant in PyTorch’s computational graph. During the gradient calculation, it reduces to zero, rendering the entire training process meaningless.
6.	In Section 2.3, the explanation of Equation (1) is unclear. The first and second conditions are mutually exclusive, meaning the third and fourth conditions will not be executed.
7.	In Section 4.2, the authors mention a test set of 261 structures, but they do not provide the specific names of these structures or explain how they were selected.

---

### Official Review · Reviewer_sHEk · 2024-11-04

**Soundness:** 3
**Presentation:** 3
**Contribution:** 2
**Rating:** 5
**Confidence:** 4

**Summary:**

The present study essentially offers a docking evaluation metric named Comprehensive Accurate Assessment (Compass), mainly focusing on the Physico-Chemical and Bioactivity (PCB) features, to advance Deep Learning (DL)-based methods. It combines two existing metrics of PoseCheck, which examines ligand strain energy, protein-ligand steric clashes, and interactions, and AA-Score, that calculates binding affinities. It is then combined and tested with an existing, recent DL-based docking method of DiffDock. An experimental analysis is reported on a well-known, commonly-used dataset of PDBBind.

**Strengths:**

+ The development of representative and comprehensive docking evaluation metrics is crucial in its own, as it has the potential to enhance the performance of most existing docking approaches without requiring significant algorithmic advancements. In this context, Compass offers a valuable contribution.

+ It is reassuring to note that this study shows some limitations of the PDBBind dataset, which has been extensively employed in numerous molecular docking research.

+ It is a practical and insightful to integrate the Compass evaluation metric into DiffDock as a state-of-the-art (SOTA) approach. This integration does not simply showcase the potential of Compass to enhance the performance of various docking methods but highlights its ability to improve specific aspects (referring to what Compass represents as the docking quality) of docking under a SOTA system.

**Weaknesses:**

- While the Compass criterion is useful, its novelty is somewhat limited as it combines two existing metrics. Similarly, the introduction of the Log Absolute Normalized - Mean Square Error (LAN-MSE) loss function, although useful, does not bring significant new insights.

- Furthermore, the combination of the LAN-MSE loss function with DiffDock's loss function in a weighted-sum manner of combining the two loss functions increases the complexity of the docking process, as the weights need to be carefully tuned for optimal performance.

- Additionally, as mentioned in Section 3.2.1, a limited number of weights and learning rates were tested. Instead of setting them manually, it may have been more efficient to utilize an existing hyper-parameter optimization tool to find the (near-)optimal weight values.

- The delivered performance improvement is questionable despite the advantages come about the Physico-Chemical and Bioactivity (PCB) features. Since, considering what is optimized through Compass, it is rather expected to see this outcome.

**Questions:**

Q: Given that Compass integrates two existing docking evaluation metrics, how representative or comprehensive is it in comparison to the broader landscape of available docking metrics? Additionally, considering that there is no perfect docking evaluation criterion, what are the limitations or drawbacks of Compass?

---

### Note · Authors · 2024-11-28

**Comment:**

Dear Reviewers,

We would like to express our sincere gratitude for the time and effort you invested in reviewing our submission. After careful consideration, we have decided to withdraw our paper from the ICLR 2025 conference. This decision was not taken lightly, and we greatly value the constructive feedback provided, which will undoubtedly help improve the work for future iterations.

Thank you again for your insights and support.

Best regards,

Authors #8171

**Withdrawal Confirmation:**

I have read and agree with the venue's withdrawal policy on behalf of myself and my co-authors.